# Fast custom wavelet analysis technique for single molecule detection and identification

Vahid Ganjalizadeh [1], Gopikrishnan G. Meena [1], Thomas A. Wall[2], Matthew A. Stott[2], Aaron R. Hawkins[2] & Holger Schmidt[1✉]

Many sensors operate by detecting and identifying individual events in a time-dependent signal which is challenging if signals are weak and background noise is present. We introduce a powerful, fast, and robust signal analysis technique based on a massively parallel continuous wavelet transform (CWT) algorithm. The superiority of this approach is demonstrated with fluorescence signals from a chip-based, optofluidic single particle sensor. The technique is more accurate than simple peak-finding algorithms and several orders of magnitude faster than existing CWT methods, allowing for real-time data analysis during sensing for the first time. Performance is further increased by applying a custom wavelet to multi-peak signals as demonstrated using amplification-free detection of single bacterial DNAs. A 4x increase in detection rate, a 6x improved error rate, and the ability for extraction of experimental parameters are demonstrated. This cluster-based CWT analysis will enable high-performance, real-time sensing when signal-to-noise is hardware limited, for instance with low-cost sensors in point of care environments.

[1] School of Engineering, University of California, Santa Cruz, 1156 High Street, Santa Cruz, CA 95064, USA. [2] Electrical and Computer Engineering Department, Brigham Young University, Provo, UT 84602, USA. ✉email: hschmidt@soe.ucsc.edu

The detection and identification of events in a time series is the basis of countless sensors across all fields of science and engineering. This task can often entail dealing with low-quality signals, often in the presence of considerable background noise, making it difficult to rely on simple thresholding algorithms where a signal above a set value is used to detect and classify an event. This can be addressed by improving the sensing instrument itself, but this approach is at odds with minimizing cost and complexity. An attractive alternative, therefore, is to extract the most information from the available time-domain signals by optimizing their analysis using appropriate signal processing techniques.

Among the many different sensor types that produce time-dependent signals, optical sensors are particularly representative examples. They are ubiquitous and play an ever-increasing role in biomedical applications such as molecular diagnostics, cell analysis, health and wellness monitoring, and more[1,2]. They are incredibly diverse, relying on many different phenomena for signal generation such as fluorescence, scattering, absorption, or resonance shifts[3–6]. Many of these sensors generate, a time-dependent sequence of individual events that arise from the detection of individual targets, such as cells, biomolecules, or reporter particles, that are moving or being scanned over. This scenario has significantly grown in importance recently as biosensor sensitivity has been improved with optofluidic techniques all the way to detection of single molecular biomarkers, ushering in the era of digital counting of different target types with a universal detection device. Examples include chip-scale flow cytometers[7], DNA sequencing with zero-mode waveguides[8], and direct detection of single viruses and nucleic acids using liquid-core waveguides[9,10]. Optofluidic waveguide devices have recently received significant attention due to their capability of amplification-free detection of single biomarkers such as nucleic acids and proteins[9,11,12].

These and other optical lab-on-chip approaches have opened the door to the development of compact tests for infectious disease diagnosis in the field and at the point of care, capabilities that are urgently needed as the global SARS-CoV-2 pandemic has resoundingly shown. Point-of-care devices, however, impose severe demands on the biosensor as they need to be inexpensive, compact, and robust. Therefore, they represent a particularly good example of the need to optimize signal processing techniques to deal with sub-optimal data. In addition, it is desirable to carry out this analysis in real-time which adds further demands on the signal processing algorithm. Unlike recent studies involving supervised machine-learning techniques for event detection and classification[13–15], we employ an unsupervised early event detection approach for immediate identification of events.

Here, we introduce a flexible signal processing algorithm for unsupervised detection and identification of single-particle signals. It is based on highly parallel, multi-scale continuous wavelet transform (CWT) analysis and meets the challenges for point-of-care sensors described above, specifically speed, accuracy, and sensitivity to low signal levels. The high accuracy, linear ($O(N)$) complexity, and high speed of the proposed algorithm are ideally suited for real-time event detection applications. We demonstrate the capabilities of this Parallel Cluster Wavelet Analysis (PCWA) algorithm for both single- and multi-spot excitation signals and validate it with a demonstration of single bacterial DNA detection on an optofluidic waveguide chip. For single-peak signals, an increase in speed by orders of magnitude over common CWT methods is shown. For the first time, this allows for real-time operation of the sensor which is a critical advance towards use in the field and at the point of care. Equally strong performance for other sensor types is also shown in the Supplementary Material with the example of an electrical single-molecule nanopore

sensor. For multi-peak signals, custom-designed wavelets are introduced that enable an over 4x increase in single-molecule detection rate and 6x reduction in errors compared to previously used techniques for periodic signals. In addition, this PCWA method allows for real-time extraction of additional experimental parameters such as the flow velocity of the sample liquid and its dynamic evolution. Finally, the multi-scale is suitable for further expansion by exploiting supervised machine-learning techniques toward extremely accurate multiplex detection.

## Results

**Single-particle fluorescence detection**. We emphasize that the technique to be introduced below can be applied to any time-dependent signal, irrespective of its nature (optical, electrical, etc.), transduction mechanism, or the device platform on which it is created. However, we have found that it is particularly attractive for a liquid-core waveguide optofluidic platform on which multiplexed detection of single nucleic acids, proteins, and viral particles have been demonstrated at clinically relevant concentration levels[9,12,16]. The devices are based on intersecting solid- and hollow-core antiresonant reflecting optical waveguides (ARROW) built with a foundry compatible fabrication process[17]. Figure 1a shows a schematic view of the experimental setup used in this study. The device under test consists of a $5\,\mu m \times 12\,\mu m$ microchannel terminated by fluidic reservoirs through which the target particles are driven by applied pressure. These are optically excited by orthogonally intersecting solid-core waveguides into which light from an external laser source is coupled as shown. The chip layout features two excitation options: single-spot via a single-mode (SM) waveguide or multi-spot excitation (MSE) with a multi-mode interference (MMI) waveguide. These different excitation patterns are visualized by the colored patterns in the top-down SEM image of the excitation region when the channel was filled with quantum dots in DI water (Fig. 1b). This high concentration solution creates a static image of the excitation patterns through which a single target particle move in an actual experiment. Single-spot excitation can be used for ultrasensitive singleplex detection as demonstrated by amplification-free detection of Ebola virus RNA[9]. MMI waveguides, on the other hand, create spectrally and spatially varying excitation patterns which has been successfully used for multiplexed single virus and antigen detection[12,18,19].

Figure 1c shows two examples of the signals for single-spot and multi-spot excitation generated by SM and MMI waveguides. They are produced by fluorescent nanobeads and labeled bacterial DNAs (Klebsiella pneumoniae carbapenemase), respectively. Clearly, the spatial excitation patterns are replicated in the time-domain signal and can encode additional information for each event. When the multi-spot signal is analyzed by a shift-multiply algorithm (see below), a 50,000x SNR enhancement was demonstrated[20]. These signals need to be detected and identified, which is challenging when the background is high or the signal level is low if the target is not bright or fluorescence collection is reduced depending on the particle's position in the channel[21]. Therefore, an efficient, powerful, and accurate signal analysis method is needed.

**Wavelet analysis for time-dependent signals**. Wavelet analysis is a well-established technique that has been successfully applied to a broad spectrum of applications, including but not limited to denoising, baseline removal, and spike detection in noisy signals (23–27). Among wavelet transform families, both continuous wavelet transform (CWT) and discrete wavelet transform (DWT) have successfully been used in multi-scale peak and event detection. DWT efficiently decomposes a sampled signal into

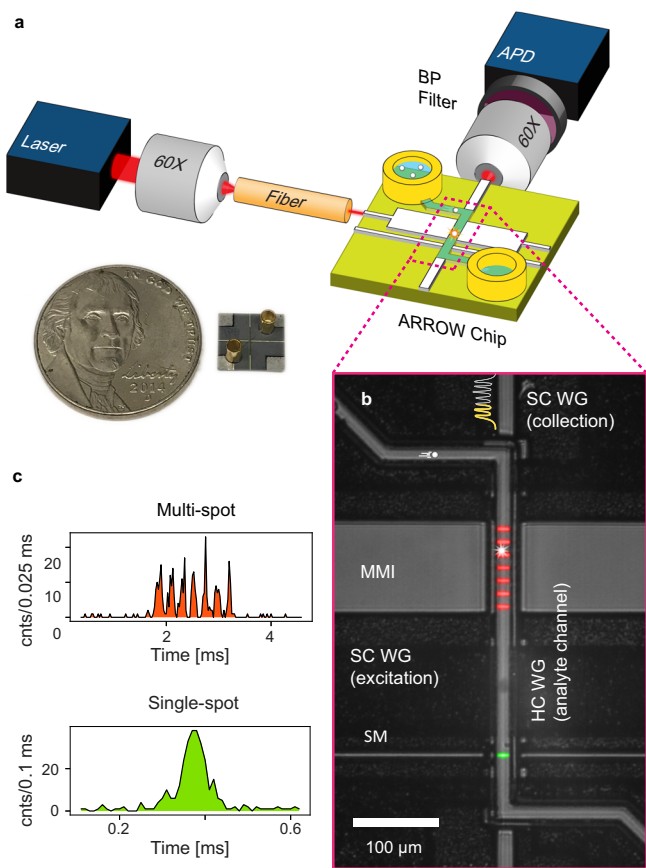

**Fig. 1 Optofluidic single-particle detection platform. a** Schematic of the experimental setup with a photo of comparison of fabricated ARROW chip with a nickel coin. The chip layout offers two options: single-peak or multi-peak fluorescence signals by coupling optical fiber into single-mode or multi-mode interference (MMI) waveguides, respectively. **b** Microscope image of detection region excited at 556 nm in the single-mode waveguide and 633 nm in the MMI waveguide which generates seven distinct spots in the analyte channel (analyte channel contains quantum dot-DI water solution for visualization of the excitation patterns as indicated by the red and green overlay colors). An example of target particles flowing across the MMI pattern and creating an intensity-modulated temporal signal in the collection waveguide is shown. **c** Examples of generated fluorescence signals: Single-peak signal is taken from fluorescent nanobeads and the multi-peak signal is generated by a single fluorescently tagged nucleic acid from a Klebsiella pneumoniae carbapenemase (KPC) bacterium. Areas under curves are shaded with the same color used in the overlay for visualization purposes.

which can be visualized in a 2D map of similarity coefficients of the signal $f$ with $\psi$ at time $t$ and scale $s$. Local maxima in the 2D map indicate the presence of a particular pattern (similar to the mother wavelet) and by the use of the scaling factor, we look for this pattern at a broad range of scales. This turns out to be the ideal unsupervised method to match patterns as the ones shown in Fig. 1c without any extra training steps (collecting and labeling training dataset). We note that the convolution calculation at each scale can be done independently for CWT whereas in DWT higher level coefficients depend on the lower level's values. The multi-scale nature of CWT analysis in the $(t,s)$ plane is particularly advantageous because events are commonly found at various $s$ values which can provide an additional source of information. In our optofluidic devices, for example, particles flow at different speeds and create fluorescence signals of different temporal widths $\Delta t$. It is, therefore, convenient to relabel the scaling parameter $s$ as $\Delta t$ to extract meaningful information for events which can, in turn, be easily converted into the velocity of flowing particles due to the direct correspondence with the known spatial excitation patterns. This is a key advantage over DWT, whose limited number of scale levels is not sufficient to extract the continuous distribution of the velocity of flowing particles.

Consequently, CWT seems ideally suited for analyzing single-peak signals such as in Fig. 1c. Indeed, the technique has been used in numerous applications such as mass spectroscopy[22], powder x-ray diffraction[23], seizure detection from EEG signal[24,25], radar target detection[26] and trend detection, and estimation in hydrology and climate research[27]. However, a major challenge lies in dealing with large amounts of data in a fast and memory-efficient way as well as with more complex signal shapes such as the multi-peak signal of Fig. 1c. We introduce a fast CWT event detection algorithm, that addresses these problems, using the example of single-peak detection.

**Parallel cluster wavelet analysis (PCWA) for single-peak detection.** Figure 2a shows a typical time-dependent signal for single-particle detection with the optofluidic chip of Fig. 1 used with the single-mode waveguide for excitation. A solution containing 0.1 pM of 200 nm polystyrene beads (Fluospheres™) were pulled through the analyte channel by connecting a vacuum line to the outlet reservoir. The photon events corresponding to the fluorescence emission from the beads were collected off the chip in 10 μs bins for further analysis and, therefore, large amounts of data points are acquired rapidly. The two-second long trace displayed in Fig. 2a contains 200,000 points, and a 5 min acquisition time produces over 30 million points. Each spike corresponds to a single nanobead. The signal height and width vary due to different particle positions in the fluidic channel[21] and fluctuations in flow speed, respectively.

The most straightforward way to detect the particles is to define a threshold of photon counts above the background and to count each crossing of this threshold as a particle[9]. While this works reasonably well, CWT analysis offers significant advantages in terms of accuracy, robustness, and information content.

The 2D CWT transform of the fluorescence trace according to Eq. (1) was computed using a Ricker wavelet (also known as Mexican hat wavelet) and is displayed as a color map in Fig. 2b. Fluorescence signals in the real-time trace are now represented as bright streaks, and the $\Delta t$ locations with the largest CWT coefficient correspond to the actual events and are highlighted with white boxes. The challenge lies in carrying out this assignment correctly and efficiently. To illustrate how this can be accomplished with a new, cluster-based algorithm, we take a look at a zoomed-in segment where a few events are displayed both in real-time (Fig. 2c) and in the $C(t, \Delta t)$ plane (Fig. 2d).

nonoverlapping sub-bands of frequencies. DWT is generally fast and efficient but lacks a sufficiently high resolution of scale/frequency. CWT, on the other hand, provides high scale/frequency resolution which is one of the key pieces of information used in time-frequency analysis. CWT is based on comparing the signal $f(t)$ to a temporal pattern of finite duration—the mother wavelet $\psi(t)$ —and identifying occurrences of the mother wavelet pattern as real events. Mathematically, this is implemented by the CWT function described by

$$C(t,s) = \left\langle f, \psi_{t,s} \right\rangle = \int_{-}^{+} f(t') \frac{1}{\sqrt{s}} \psi^* \left( \frac{t-t'}{s} \right) dt', \quad (1)$$

where $s > 0$ is a scaling factor that effectively stretches or compresses the wavelet pattern in time. $C(t, s)$ is, therefore, the correlation of the real signal with a scaled and dilated basis function

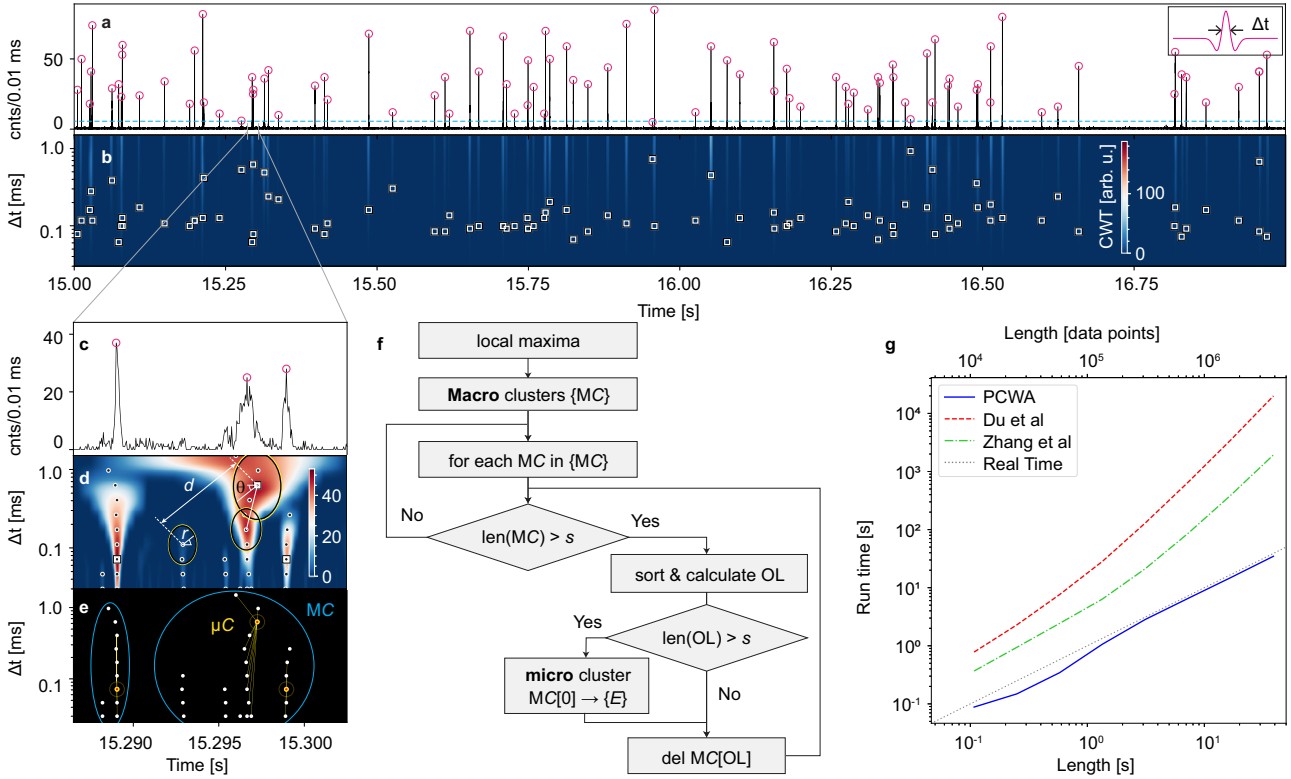

**Fig. 2 Parallel cluster wavelet analysis (PCWA) for single-peak analysis. a** Cropped window of fluorescence signal taken from 200 nm fluorescent beads excited by single-mode (SM) waveguide (inset: Ricker wavelet used with PCWA algorithm). **b** CWT coefficients in time-$\Delta t$ space (where a scaled and dilated version of the mother wavelet is convolved with raw data) with square markers indicating selected local maxima points found by the PCWA event detector algorithm. **c** Zoomed-in window of three events with circle markers showing the adjusted location of peaks. **d** CWT map of (**c**) including local maxima points (black dots). The clustering algorithm utilizes Euclidean distance and adjusted ellipses around each local maximum to search for links. The overlap of an ellipse with the centroid point defines a link. **e** Macro and micro clusters (MC and μC): local maxima are first grouped into MC highlighted by blue circles by simplified 1D overlap calculation. The clustering algorithm finds μC for each MC in parallel. A μC is a star graph containing a minimum of links with the largest CWT coefficient maximum as the centroid (red-filled circles). **f** Flowchart of the clustering algorithm. **g** Run time comparison of clustering algorithm with established CWT peak finders, showing orders of magnitude faster speed and run times below the real-time limit (gray dashed line).

First, local maxima (black dots in Fig. 2d) in the CWT map are identified at each scale ($\Delta t$ value) with a conventional peak-finding (with a user-defined threshold) process. Previously developed CWT algorithms now select the correct points using a ridge-line approach starting from the maxima at the largest $\Delta t$ value[22,28]. In the most direct approach[22], ridges of connected points are developed sequentially by identifying maxima in the adjacent $\Delta t$ row that are within a predefined distance in the CWT map. Once a ridge is formed, the time location of the filtered ridges with an SNR value above a threshold is identified as the event location. Since the algorithm is serial in nature (top to bottom), it is computationally intensive and slow. It is also not accurate to locate event locations. A refined version of this method proceeds in the same way[28], but uses additional ridges that are calculated from zero crossings and local minima in the CWT map for better identification and extraction of additional information (peak width). Due to the additional information used, this method is more accurate than the direct ridge analysis, but it is also even more memory intensive.

In order to enable real-time CWT analysis, we introduce a highly tunable, faster, and more memory-efficient algorithm based on cluster analysis. The flowchart for this process called Parallel Cluster Wavelet Analysis (PCWA) is depicted in Fig. 2f. In contrast to previous algorithms, we first define macro clusters (MC) of CWT maxima that are separated by gaps along the $t$-axis by more than a predefined value. We then examine all clusters in parallel by calculating the distance values around the local maximum with the largest $C(t, \Delta t)$ value which is the most likely candidate for an event. This, too, is done in a parallel, vectorized fashion, making the algorithm both efficient and fast. The overlap OL with other maxima within the cluster is determined from

$$\text{OL}(i, 0) = \text{sgn}\left(\left(r_i + r_0\right)^2 - d^2(i, 0)\right)$$

$$d^2(i, 0) = \left(t_i - t_0\right)^2 + \left(\Delta t_i - \Delta t_0\right)^2$$

$$r_i = \frac{whN_i \Delta t_i \sqrt{C_i'}}{\sqrt{w^2 N_i^2 \sin^2\theta_i + h^2\cos^2\theta_i}}, C_i' = \frac{C_i - \min(C)}{\max(C) - \min(C)}, r_0 = \frac{whN_0 \Delta t_0}{\sqrt{w^2 N_i^2 \sin^2\theta_i + h^2\cos^2\theta_i}}$$

(2)

where $r$ is the radius of the ellipses in Fig. 2d, $d$ is the Euclidean distance, $w$ and $h$ are adjustable spreading parameters that define the refinement sensitivity in time and scale, respectively. $N$ represents the number of peaks in a multi-peak signal (here equal to one). The addition of normalized CWT coefficient weights into $(r_i + r_0)^2$ helps detect weak events near strong ones.

We then look for overlapping ellipses. If the number of points connected to the original largest maximum (centroid) is higher than a user-defined number, it is taken as a micro cluster (μC) in which the actual event is immediately identified. The unconnected points form a new, smaller macro cluster where a new centroid is picked. The analysis repeats until no more clusters can be formed with a minimal, user-defined number of candidate points.

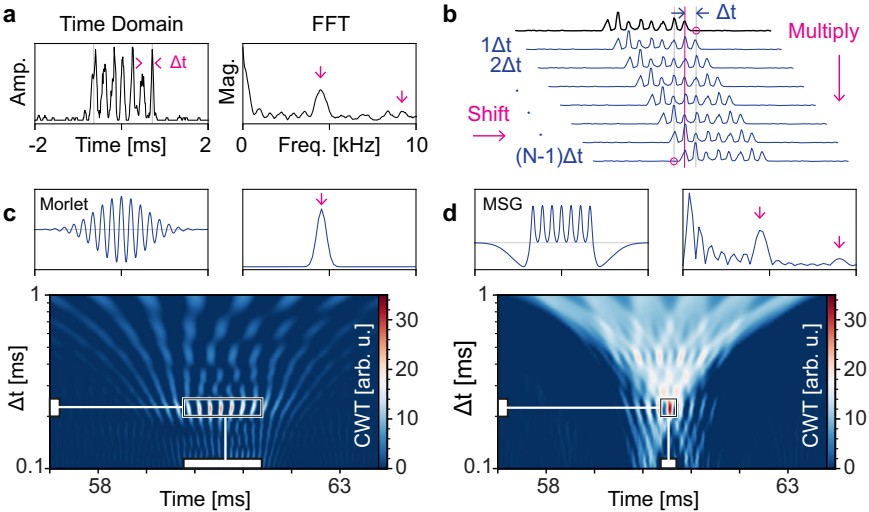

**Fig. 3 Multi-peak detection methods. a** Time and frequency domain representation of a fluorescence signal taken from single K. pneumoniae Carbapenemase DNA molecules excited by seven excitation spots from the multi-mode interference (MMI) waveguide. Arrows in FFT indicate the first and second harmonic of the time trace. **b** Shift-multiply algorithm previously used to find and classify multi-peak signals. **c** Time and frequency information of Morlet wavelet with single component aligned to the first harmonic of multi-peak signal. 2D map of positive CWT coefficients with white boxes indicating approximate certainty of time and scale localization. **d** Multi-spot Gaussian (MSG) wavelet designed to match the multi-peak signal as well as the full frequency spectrum; CWT coefficients map shows precise event localization.

Figure 2b showed the detection of fluorescent nanobeads using this PCWA algorithm and suggests high accuracy. Indeed, when compared to a conventional amplitude-based peak finder, 5.7% more events were detected. In addition, the particles' velocity is also extracted from the scale values which correspond to the temporal widths of the fluorescence peaks. This is discussed in more detail in the next section, but the analysis for these single-peak signals is also provided in the supplementary information (Supplementary Fig. S1).

Figure 2g compares the run time of the PCWA algorithm with the ridge-based CWT methods as a function of both the number of data points in the signal trace and the actual run time of the experiment. The comparison was done by running implementations of the algorithms in Python with 100 logarithmic scale values on a single desktop computer (Intel® Core™ i9-900 CPU with 32 GB of RAM). We find that our method has $O(N)$ complexity, where $N$ is the number of data points. Consequently, it is orders of magnitude faster than other techniques and, most notably, our clustered CWT analysis is always faster than the experiment itself (dotted line), e.g., 40 s of analysis for a 42 s trace, while other methods always take orders of magnitude longer than the experiment and quickly become impractical. Consequently, PCWA enables real-time analysis of time-dependent particle sensors. This is the first principal result of this work.

We note again that the PCWA method is widely applicable to other sensor types. The example of single DNA detection with nanopore electrical sensor chips is presented in the supplementary information and confirms the performance improvements described in Fig. 2. Another important question concerns the accuracy of the PCWA method which can be assessed with the help of a receiver operating characteristic (ROC) curve that considers both true and false-positive events. In order to carry out a ROC curve analysis, a dataset with known positive events is required. We used a set of 75 mass spectrometry traces from a simulated protein spectra dataset[29] and found that our algorithms slightly outperforms other CWT techniques. Details are provided in the supplementary information (see Supplementary Fig. S3). This shows that our cluster CWT algorithm is fast, efficient, accurate, and can be applied to a diverse range of sensor types.

**Clustered CWT analysis of multi-peak signals**. Multi-peak signals offer significant advantages for practical sensing applications. By introducing redundancy and patterning to the signal, more reliable identification of events from a noisy background that resembles single peaks becomes possible. Multi-peak signals also enable multiplex detection if different targets produce different signal patterns which is particularly desirable for biomedical applications. In practice, this often involves the use of spatial excitation patterns using masks[30] or waveguides[20,31] that result in a corresponding time modulation of the signal created by a particle that passes by such a pattern. These can then be analyzed using Fourier transform analysis, shift-multiply algorithms (see below)[20], or matched filters. For example, signal-to-noise ratio (SNR) enhancement via multi-spot excitation implemented with Y-splitters and multi-mode interference (MMI) waveguides has shown up to 50,000x SNR improvement[20,31].

Improved signal analysis with spatially patterned excitation has been demonstrated in various implementations[32–34]. To demonstrate the benefits of the PCWA approach in this context, we analyze signals from multi-spot excitation of individual fluorescently tagged plasmid molecules corresponding to the Klebsiella pneumonia carbapenemase (KPC) bacterial species. The multi-spot excitation pattern is generated by a multi-mode-interference (MMI) waveguide[18,35]. In this experiment, a HeNe laser source was fiber-coupled into the MMI waveguide (see Fig. 1a) while the rest of the setup was identical to the one used in the single-peak detection experiment. Figure 3a shows an example of a multi-peak signal from a single DNA molecule that consists of seven peaks in response to the MMI waveguide excitation pattern generated at 633 nm. Also shown is the Fourier transform of this signal which shows a strong peak at 4.5 kHz that arises from the uniform spacing $\Delta t$ of the seven signal peaks. In addition, the second harmonic at 9 kHz and strong content at very low frequencies are visible. In the past, such multi-peak signals have been detected and classified (by peak number) for multiplex detection using a shift-multiply algorithm[20,31,34] with good results. This benchmark algorithm is briefly reviewed in Fig. 3b. It is based on recursively (N-1 times) shifting a selected multi-peak event by $\Delta t$ (the uniform spacing between $N$ adjacent peaks)

and multiplying all shifted signals with each other. The resulting product is large for a correct signal and very small for incorrect peak numbers or temporal spacing, resulting in significant SNR improvement compared to purely threshold-based counting[20,31].

We now demonstrate dramatic improvements using PCWA with custom-designed wavelets. First, we choose another mother wavelet that is commonly used in CWT data analysis. The Morlet wavelet (Fig. 3c, top left) contains frequency information embedded by enveloping a sinusoidal (real Morlet) or exponential (complex Morlet) function with a Gabor window (Gaussian function). The frequency spectrum (FFT) of the Morlet wavelet (Fig. 3c, top right) correctly reproduces the main feature of the fluorescence signal (the first harmonic) but does not show the finer features of its spectral content at higher and lower frequencies.

Figure 3c (bottom) shows the CWT map when the Morlet wavelet is applied to the signal of Fig. 3a. Multiple red spots with large CWT coefficients are seen at times where the Morlet wavelet peaks line up with the peaks in the fluorescence signal. This indicates that particles can indeed be detected, but with some temporal uncertainty for the event as highlighted by the white boxes on the time and scale axes.

An even better strategy is to construct a mother wavelet that is custom-designed for the signal pattern to be detected, here the 7-peak MMI signal of Fig. 3a. To this end, we define a Multi-Spot Gaussian (MSG) wavelet as seen in Fig. 3d (top, left). It is constructed by the sum of $N$ Gaussians separated by $\Delta t$ and surrounded by two negative skewed peaks and mathematically described by

$$\psi_N(t, \Delta t) = \sum_{n=0}^{N-1} \exp\left(\frac{-\left[t - \left(n - \frac{N-1}{2}\right)\Delta t\right]^2}{2\Delta t^2 \sigma_+^2}\right)$$
$$- \sum_{k=\pm 1} \frac{2a}{\Delta t \sigma_-} \phi\left(\frac{t + k(\sigma_- m_0 - \frac{N}{2})\Delta t}{\Delta t \sigma_-}\right) \Phi\left(k\alpha \frac{t + k(\sigma_- m_0 - \frac{N}{2})\Delta t}{\Delta t \sigma_-}\right),$$

(3)

here, the $\sigma_+$ parameter for positive peaks is fitted to those measured from multi-peak signals normalized to $\Delta t$. $a$ and $\sigma_-$ are calculated according to positive peaks to achieve optimal compactness and sensitivity. For higher sensitivity of the wavelet to $N$, the maxima ($m_0$) of the skewed Gaussian functions at both ends are placed at $\Delta t$ from the first and last positive peaks. The negative side peaks of the wavelet ensure that the zero mean condition applies[36], and the wavelet is scaled for a square norm of one. The parameters of the skewed Gaussian functions are explained in the supplementary material.

The Fourier transform of the MSG wavelet is shown in Fig. 3d (top, right) and shows excellent qualitative agreement with the FFT of the multi-peak signal from the KPC target. Figure 3d (bottom) shows the CWT map obtained with the custom MSG mother wavelet in Eq. (3). It shows a single dominant bright spot (black box) that demonstrates clear particle detection and excellent localization on both the time and $\Delta t$ axes (white boxes), corresponding to precise determination of the particle detection event and its velocity.

We then applied the PCWA method with the MSG mother wavelet to a full, 20-min-long experimental trace of fluorescently tagged DNA molecules detection. Figure 4a shows a snapshot of this trace with clean identification of seven molecules, each producing a seven-peak signal. The CWT map (Fig. 4b) shows clean identification of these signals and also illustrates the variation in $\Delta t$ due to the different velocities of the molecules. Note that the gaps in the time axis of Fig. 4a, b were added solely to enable visualization of multiple plasmid detection events, which are very sparse at low concentrations.

We note that conventional ridge-based CWT methods cannot be applied to multi-peak signals. This is illustrated in Fig. 4c where the local maxima (equivalent to Fig. 2c) around a single seven-peak event are shown. Clearly, this single event results in multiple maxima at each $\Delta t$ level, creating multiple ridges. Consequently, ridge-based identification would detect multiple events and fails. In contrast, our PCWA algorithm recognizes all of these points in a single step as connected to the largest maximum in the group. As a result, the event forms a single micro cluster that is quickly identified correctly. Like in the case of single-peak signals, the analysis time remains shorter than the run time of the experiment. The unique ability of the PCWA algorithm to both detect and identify multi-peak signals using custom wavelets represents the second principal result of this manuscript.

In order to compare the performance of the three analysis approaches (Shift-Multiply, PCWA (Morlet), and PCWA (MSG), we evaluated both the measured single-molecule detection rates and their accuracy. Here, accuracy was defined by the algorithms' ability to identify the correct peak number (here: seven) of a detected event. This is meaningful because multiplexing with MMI waveguides can be implemented by simultaneously generating signals with different peak numbers using the spectral and/or spatial dependence of the MMI pattern[18,35]. Therefore, the three methods were applied to each event with three different peak numbers (6, 7, 8). For example, $C(u,s)$ was determined with three different MSG wavelets and the coefficient with the highest value was chosen for each event. For each method, a series of threshold values were scanned to find the optimal compromise between accuracy and detection rate. The results of this comparison are presented in Table 1 and clearly show that PCWA analysis with the custom MSG mother wavelet offers by far the best performance with over 4x more identified events and 6x fewer errors than the shift-multiply algorithm.

In order to verify the superiority of PCWA over Shift-Multiply and MSG over Morlet wavelet, we did additional analysis done a simulated multi-peak signal (Supplementary Fig. S4) with a known ground truth events list. The results (see Supplementary Fig. S5 and Table S1) show a close match with the real-world results shown in Table 1.

Finally, we demonstrate the ability of the PCWA method to instantly extract additional valuable information from the sensor data. Because the optical excitation patterns are generated by lithographically defined waveguides according to the MMI principle, they have a well-defined physical spacing. Therefore, the $\Delta t$ value of an event identified in the CWT map, which represents the temporal spacing between peaks in the fluorescence signal, can directly be converted into a flow velocity for the particle. This is visualized in Fig. 4b by the velocity axis on the right.

In Fig. 4d, the intensity and $\Delta t$ (velocity) of all 300 detected DNA molecules are visualized in a 2D histogram to provide joint information analysis. The events are distributed in a pattern that is determined by the waveguide mode patterns and the (parabolic) velocity profile in the microchannel. The predicted distribution of the majority of events for the chip under consideration is shown as a white line and matches the data well. A few noise peaks incorrectly identified by the CWT algorithm as fluorescent particles lie outside the white dashed ellipse and can now be rejected based on the additional velocity information. Figure 4e shows the dynamics of speed and intensity of particles inside the fluidic channel over the course of the experiment. Good agreement of velocity fluctuations and detection rate is observed. Real-time analysis can be easily implemented by running the event detection algorithm on a sliding window of the buffered signal during data acquisition.

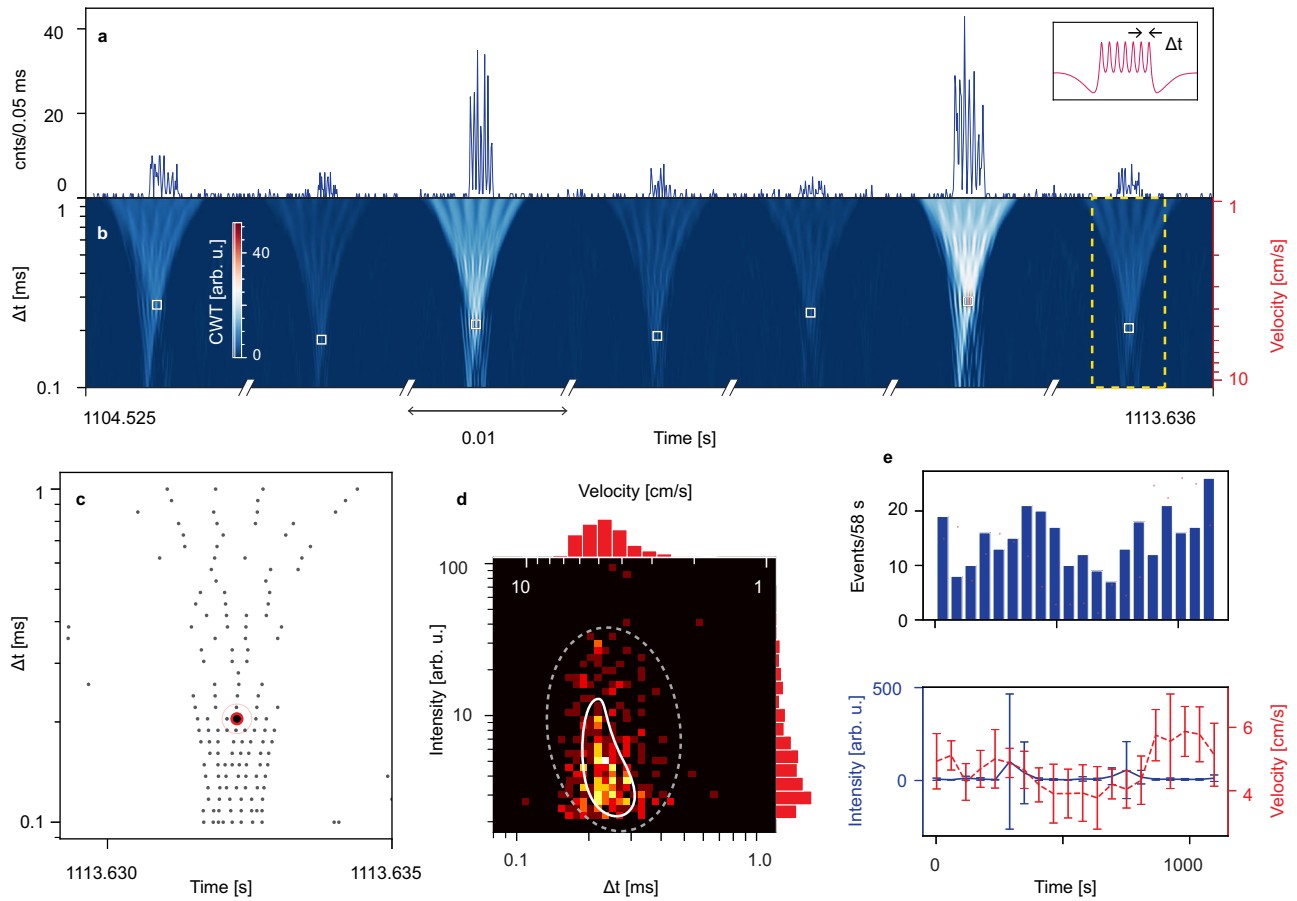

**Fig. 4 Multi-peak analysis. a** Concatenated 10 ms cuts of detected single KPC molecules in a 9 s window. Inset is the multi-spot Gaussian (MSG) wavelet used to analyze the trace. **b** Corresponding time-$\Delta t$ CWT scalogram with white squares showing detected multi-peak events across a range of $\Delta t$ values. **c** Local maxima for a single example event render conventional ridge-line methods impractical. **d** Scatter plot for particle intensity and speed, showing a cluster in the predicted region (white line); dashed line: confidence region for event identification as DNA molecules. **e** Time-varying information of events during the measurement for dynamic determination of the flow characteristics. Intensity and velocity plots are the average value for events within a bin from the histogram plot. Error bars represent standard deviation.

**Table 1 Performance comparison for single-molecule KPC analysis using different algorithms.**

|  | Shift-multiply | PCWA (Morlet) | PCWA (MSG) |
|---|---|---|---|
| Detection rate [x10⁴ events/mL] | 1.03 | 1.84 | 4.60 |
| Accuracy [%] | 66.37 | 37.04 | 94.03 |

Such real-time statistics can help monitor proper experimental operation, e.g., maintaining constant vacuum pressure.

## Discussion

We have introduced a Parallel Cluster Wavelet Analysis (PCWA) algorithm that operates in a highly parallel fashion, enabling fast, accurate, memory-efficient detection, and identification of events in real-time. The algorithm's characteristics are ideal for single-particle sensors that produce large amounts of data, possibly over long periods of time. The PCWA principle was validated on a diverse range of data sets, in particular, a chip-based, optical single biomolecule detection assay that is compatible with diagnostic applications and settings with low signal-to-noise conditions such as point-of-care use. Fluorescence signals from single particles were rapidly and fully analyzed via a single transform operation which also enabled the determination of the velocity and intensity of the individual molecules. The use of multi-peak signals with a customized mother wavelet outperforms other established detection methods. This combination is especially attractive for multiplex detection where different classes of targets generate signals with a different number of peaks. In the future, the technique can be further expanded to dynamically (on-the-fly) adapt the shape of the multi-peak mother wavelets to the patterns generated by a particular sensor to further improve the performance. This flexibility would be ideal for point-of-care devices because it further alleviates the demands on the tolerances and cost associated with the device fabrication process to produce identical excitation patterns. Moreover, the algorithm can also be applied to automated dataset collection and labeling for super-vised machine-learning tasks applications.

## Methods

**Optofluidic chip fabrication**. Optofluidic chips were fabricated by depositing alternating layers of high ($Ta_2O_5$) and low ($SiO_2$) refractive index on top of an <100> oriented silicon substrate, forming the ARROW layer to guide the light in solid core and a liquid core. Waveguides were then built on top of the ARROW layer stack using low-stress PECVD and a sacrificial layer process[17]. The present

implementation uses a dual-oxide buried ARROW design with optimized lateral confinement and performance[37]

**Sample preparation**. The KPC sample used in the multi-peak signal experiment was prepared by mixing a 1 μM concentration of cell-permeant SYTO 62 red-fluorescent nucleic acid stain (Thermo Fisher Scientific) with a KPC plasmid sample prior to detection. SYTO 62 dyes fluoresce when bound to DNA, making it possible to detect and count individual nucleic acids without amplification[11].

**Experimental setup**. A single-molecule detection setup was built to excite fluorescent-tagged target particles and collect and record the emission from targets. A solid-state diode neodymium-doped yttrium aluminum garnet; $Nd:Y_3Al_5O_{12}$ (SSD Nd:YAG) laser (Shanghai Dream Laser Technology Co.) and a HeNe laser (Melles Griot) working at 556 and 633 nm, respectively, were coupled into a single-mode fiber using a 60x microscope objective lens (Newport). The single-mode fiber was then butt-coupled into the optofluidic chip. The fluorescence emission from tagged particles was coupled into the liquid-core ARROW waveguide and then guided off the chip via a solid-core waveguide. These photons were collected by a 60x microscope objective lens (Newport), passed through a penta-bandpass optical filter (FF01- 440/521/607/694/809–25, Semrock) to remove excitation light, and detected by an avalanche photodiode (APD, Excelitas). Single-photon events were recorded with a photon-counting card (Picoquant, TimeHarp 260 Nano) into a desktop PC for further analysis.

## Data availability

The raw fluorescence data are available under restricted access for data privacy reason. Access can be obtained by a reasonable request from the corresponding author.

## Code availability

The source code of the proposed PCWA algorithm with example scripts are available at https://github.com/vganjali/PCWA [38].

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

## Acknowledgements

We acknowledge R.L. Hanson, R.L. Wood, O.T. Brown, R.A. Robison, W.G. Pitt, and A.T. Wooley for the preparation of the KPC samples and M.J.N. Sampad for providing the date for the nanopore experiment discussed in the SI. This work was supported by a gift from the Cisco University Research Program Fund (gift # 2020-224954, H.S.) as well as the National Institutes of Health under grants 1R01AI116989-01 (A.R.H.), 1R01EB028608 (H.S.), and National Science Foundation under grant CBET-1703058 (H.S.), respectively.

## Author contributions

V.G., A.R.H., and H.S. designed the research; T.A.W. and M.A.S fabricated the optofluidic devices; V.G and G.G.M. performed the experiments; V.G. developed the PCWA algorithm, V.G. and H.S. performed data analysis; V.G., G.G.M, A.R.H., and H.S. wrote the paper.

## Competing interests

A.R.H. and H.S. have a financial interest in Fluxus Inc. which commercializes optofluidic technology. The remaining authors declare no competing interests.
