## [Peer Review File · Nature Communications]

REVIEWER COMMENTS

Reviewer #1 (Remarks to the Author):

The authors propose " a new Parallel Cluster Wavelet Analysis (PCWA) algorithm that operates in highly parallel fashion, enabling fast, accurate, memory-efficient detection and identification of events". But they did not propose any quantitative description of how fast the algorithm is.

Is it an Order(N) or $O(N\log N)$ algorithm? (N represents the number of data points).

If the data is regularly sampled, the authors can use a fast discrete wavelet transform (DWT) algorithm to detect events. If yes, they should compare their method with a fast DWT. Since there is no reconstruction requirement in the proposed event detection framework they can use a fast DWT.

Reviewer #2 (Remarks to the Author):

In this paper the authors present a "robust signal analysis technique" using a parallel continuous wavelet transform algorithm and demonstrate it using a single particle microfluidic sensor. The technique is proposed to be more accurate than traditional methods and faster than other continuous wavelet transform techniques. The team demonstrates a 4x increase in detection rate, what they state is a 6x improved error rate, and processing on a "real-time" timescale.

I think the paper is interesting, complete, and conceivably of broad interest. I think it could be publishable in Nature Communications subject to the following comments.

Overall my main comment is that I don't quite understand the fundamental innovation that has been introduced that has enabled the improvement. I have some comments about the innovation below, but there does appear to be significant advancements over the state of the art presented. As I read the very long paper however these advancements appear to come from a set of evolutionary enhancements to the existing algorithms. This is fine of course, but for nature communications I would expect more clarity on what the more "revolutionary" advancement is. It is perhaps expressed in the details of the paper but I think the impact of the paper would be improved if it could be expressed what the major impact is as opposed to grinding through an algorithm to make it better.

Additionally the comparison (at least in terms of speed) is made against a paper from 2006 and another one that appears somewhat obscure and implements the technology for a different application. How do we know these represent the state of the art? CWTs are popular in a number of research areas – how does the improvement here compare with a broader set of algorithmic advancements? Commercial techniques?

Finally part of the increase in accuracy claimed is a reduction in the error rate – which as I read is based on more peak detection. How does one know the ground truth? How do we know that the greater number of hits aren't false positives (or at least some of them)? I assume the different algorithm has some disadvantages compared to the state of the art, what are they?

A minor, or fixable, point is that in Figure 1 (b) and (c) do not seem to match what is described in the caption. It also appears that what is shown in (c) is not the same experiment as is shown in (b) – which is implied by the image (though not the caption). One appears to be a quantum dot and the other a tagged bacteria. It would be best if the images were from the same experiment.

Response to reviewers' comments, manuscript NCOMMS-21-18849

We would like to thank the reviewers for their detailed evaluation of the manuscript. We very much appreciate their encouraging comments such as “the paper is interesting, complete, and conceivably of broad interest”.

The reviewers also have some valid queries along with effective suggestions. We highly appreciate the comments and have included necessary revisions to our manuscript.

In the following, we discuss comment-by-comment responses to the reviewers' points. All modifications in the revised manuscript and the supplementary information are highlighted in blue font.

Overview of major revisions:

The reviewers' major concern was verification of the improvements in event detection rate and accuracy achieved by our new PCWA algorithm. As suggested, we carried out extensive new simulations with ground truth data and were able to confirm our experimental findings. We have included the additional analysis in the supplementary information. The article was revised to reflect the necessary changes to address the reviewers' concerns.

Reviewer #1:

1. The authors propose " a new Parallel Cluster Wavelet Analysis (PCWA) algorithm that operates in highly parallel fashion, enabling fast, accurate, memory-efficient detection and identification of events". But they did not propose any quantitative description of how fast the algorithm is.

Response: We thank the reviewer for highlighting this. We had included a run time comparison between the different algorithms in Fig. 2g which showed quantitatively that our PCWA algorithm operates orders of magnitude faster. The run time is fast enough to run in real-time (even for larger numbers of datapoints). We had, however, not included a statement of the functional behavior of the run time and are addressing this with the next point.

2. Is it an Order(N) or $O(N\log N)$ algorithm? (N represents the number of data points).

Response: We take advantage of breaking long signals into smaller chunks and running PCWA algorithm on each chunk in parallel. Specifically, the macro-clustering step, provides an additional parallel analysis of local maxima points in the clustering step. Keeping the chunk size fixed and utilizing macro-clustering ensures that the algorithm has \$O(N)\$ complexity. This was reflected visually in Fig. 2g as the run time of PCWA was linearly increasing with the length of the signal (N). The manuscript has been revised to include this important information.

3. If the data is regularly sampled, the authors can use a fast discrete wavelet transform (DWT) algorithm to detect events. If yes, they should compare their method with a fast DWT. Since there is no reconstruction requirement in the proposed event detection framework they can use a fast DWT.

Response: We thank the reviewer for the question. We chose CWT over DWT for a few reasons that we summarize below:

- CWT provides higher scale resolution (referred to as \$\Delta t\$ in the manuscript) than DWT. Scales in DWT are incremented in steps of \$2^n\$ (down sampling by a factor of 2 in each level). For example,

in a multi-peak analysis, events have Δt values ranging from 0.1 ms to 1.0 ms (see Fig. 3 bottom). With DWT, we can only have 4 scales (0.1×2^0 , 0.1×2^1 , 0.1×2^2 , 0.1×2^3) which results in a very coarse Δt approximation, while with CWT we can have any number of scales (50 scales in Fig. 3 bottom).

- We utilize CWT to detect events as well as extract other information like the speed of a particle. This is crucial in statistical analysis of the events, such as estimating target molecule concentrations in the solution. The precise locating of the events in the scale axis is available in CWT as mentioned above.
- DWT is usually followed by a complicated supervised classifier to identify events in time series. The added training step negates the speed up attained from DWT and makes the full analysis slower than the CWT algorithm introduced here.

We have summarized this discussion in the revised manuscript.

Reviewer #2:

1. Overall my main comment is that I don't quite understand the fundamental innovation that has been introduced that has enabled the improvement. I have some comments about the innovation below, but there does appear to be significant advancements over the state of the art presented. As I read the very long paper however these advancements appear to come from a set of evolutionary enhancements to the existing algorithms. This is fine of course, but for nature communications I would expect more clarity on what the more "revolutionary" advancement is. It is perhaps expressed in the details of the paper but I think the impact of the paper would be improved if it could be expressed what the major impact is as opposed to grinding through an algorithm to make it better.

Response: We thank the reviewer for raising this critical issue. The short answer is that the fundamental innovation in our work is that our new algorithm allows for real-time analysis of sensor data. This is critical for use in the field or at the point-of-care where operational speed is of the essence and in direct contrast with existing CWT algorithms which are orders of magnitude slower. Therefore, this constitutes a significant change that enables qualitatively new analysis capabilities, and, we believe, a major advance.

In addition, the new PCWA algorithms features multiple innovative aspects and advances that we attempted to highlight throughout the manuscript. For example, unlike previous algorithms, PCWA takes advantage of time, scale and coefficient information. Next, the example of customized multi-peak signal analysis highlights the applicability of our approach to a broader range of signals and the capability for unsupervised event identification. Finally, we reported quantitative improvements attained with the novel approach of CWT-aided time-series analysis in both abstract and main text.

We realize that due to the expansive scope of the paper, there is a risk that the main innovation might have inadvertently been buried in the text. Consequently, we have revised the abstract and the summary to draw proper attention to our main point and underscore the substantial improvements we found.

2a. Additionally the comparison (at least in terms of speed) is made against a paper from 2006 and another one that appears somewhat obscure and implements the technology for a different application. How do we know these represent the state of the art?

Response: Thanks for the question. We did the comparison with those papers since they had presented the fundamental approach for locating events from CWT coefficients. Newer studies mostly involve novel supervised machine learning techniques like SVM, k-NN, etc. which have high complexity of $O(N\log N)$ or higher (see response to reviewer 1, point 2). We have added a brief discussion and references to the manuscript.

2b. CWTs are popular in a number of research areas – how does the improvement here compare with a broader set of algorithmic advancements? Commercial techniques?

Response: CWT is widely used in 1D single peak detection (i.e. mass spectroscopy, epileptic seizure, etc.) and in 2D images for denoising and feature extraction. Recent advancements in this technique are mostly based on adding machine learning techniques to the analysis. Supervised machine learning algorithms require operator involvement in creation of training datasets and additional training time. We believe our strategy of finding and identifying patterns in a 1D time series can be adapted to other applications in similar signals (recorded from other quantities) as well as in the image processing field.

3a. Finally part of the increase in accuracy claimed is a reduction in the error rate – which as I read is based on more peak detection. How does one know the ground truth?

Response: We thank the reviewer for this important question regarding one of the main advances of our work. To address this question and concern about ground truth, we developed a new set of simulations consisting of deterministically placed events that represent a known ground truth. The properties of these simulated signals (peak width, peak spacing, velocity and amplitude distributions) were modeled after those of the real signals used for Fig. 4 of the main manuscript. The figure below shows the resulting simulated data trace which was then analyzed with the same three algorithms as the real experimental data.

Figure 1: Simulated data trace with 239 known ground truth multi-peak events.

We were able to replicate the findings from the experiment, specifically a >4x higher detection rate and an improved accuracy of the PCWA algorithm compared to the shift-multiply method. In particular, our

PCWA method was able to detect 236 out of the 239 known events, much higher than the reference algorithms (49 and 189, respectively).

A full description of these simulations with additional details is presented in a new section in the supplementary material, and we have summarized it in the main manuscript. Importantly, we believe that this additional work settles the legitimate question of the reliability of our new method. We thank the reviewer again for encouraging us to carry out this important analysis.

3b. How do we know that the greater number of hits aren't false positives (or at least some of them)?

Response: We visually inspect the events to make sure they look like an event and only consider those for which we have high confidence.

3c. I assume the different algorithm has some disadvantages compared to the state of the art, what are they?

Response: We believe that, on the whole, the new PCWA algorithm provides the best, most balanced performance for a broad range of applications. It is slower than the simple Shift-Multiply algorithm as it requires more computations and prior knowledge about the shape of the multi-spot pattern in order to make the wavelet function (shift-multiply only requires the spacing/ Δt). Comparing the custom-wavelet algorithm with a standard wavelet approach, it requires an additional step of designing a custom (here: MSG) wavelet, but we believe that the significantly improved performance outweighs this additional one-time effort.

4a. A minor, or fixable, point is that in Figure 1 (b) and (c) do not seem to match what is described in the caption.

Response: Thanks for pointing that out. We have corrected the misplaced labels (b) and (c) in the figure in the revised version.

4b. It also appears that what is shown in (c) is not the same experiment as is shown in (b) – which is implied by the image (though not the caption). One appears to be a quantum dot and the other a tagged bacteria. It would be best if the images were from the same experiment.

Response: We thank the reviewer for raising this point. We use a solution of quantum dots that are homogeneously distributed at high concentration in the analyte channel in order to be able to create a static image of the excitation spot patterns for visualization with a camera from the top. In the actual experiments of Fig. 1c), single particles are detected as they flow through the channel at low concentration. These do not sample the entire channel and, therefore, do not generate an image of the complete excitation pattern. We have clarified this fact in the text and the figure caption in the revised manuscript.

Summary:

In summary, we have addressed all reviewer comments and revised the manuscript and supplementary information accordingly. We believe that the paper is now suitable for publication in *Nature Communications*.

REVIEWERS' COMMENTS

Reviewer #1 (Remarks to the Author):

- The manuscript can be accepted.
- The authors developed a continuous wavelet for their sensor/data.
- After designing the wavelet they perform "matched filtering". They do not perform wavelet analysis. In wavelet analysis, we have to change both "s" and "t" values. They perform matched filtering and they change only the "t" value. As a result, their method is "fast".

Response to reviewers' comments, manuscript NCOMMS-21-18849

We would like to thank the reviewer for evaluating our revised manuscript and recommending that “the manuscript can be accepted”.

Here, we respond to the reviewer’s final query. All modifications in the revised manuscript and the supplementary information are highlighted in blue font.

Reviewer #1:

1. After designing the wavelet they perform "matched filtering". They do not perform wavelet analysis. In wavelet analysis, we have to change both "s" and "t" values. They perform matched filtering and they change only the "t" value. As a result, their method is "fast".

Response: We appreciate the reviewer’s concern which appears to be based on a misunderstanding as we do indeed change both “s” and “t” values in our wavelet analysis. This can be seen in Fig. 2b where we are shifting the mother wavelet in “t” and scaling it in “ Δt ” when generating CWT coefficients (2D maps). We added a sentence to the caption to make it clear we are changing both of them. Furthermore, in the section “Parallel Cluster Wavelet Analysis (PCWA) for Single-Peak Detection”, third paragraph, starting with “First, local maxima (black dots in Fig. 2d) in the CWT map are identified at each scale (Δt value) ...”, we emphasize replacing “s” in conventional CWT transforms with “ Δt ” (a more informative parameter in our case) while still being used as a scaling variable (in addition to ‘t’). The remainder of that paragraph also explains why our proposed algorithm runs faster than previous algorithms.

We have confirmed that the revised manuscript is able to address reviewers concern.

Summary:

In summary, we believe that the paper is now suitable for publication in *Nature Communications*.